Designing spaces to support collaborative creativity in shared virtual environments

Men Liang l.men@qmul.ac.uk
Bryan-Kinns Nick
Bryce Louise
School of Electronic Engineering and Computer Science, Queen Mary University of London , London , United Kingdom
Carroll John
Electronic publication date: 2019 Nov 4
Publication date: 2019
Volume: 5
Electronic Location ID: e229
Received 2019 Jun 11; Accepted 2019 Sep 24
Copyright: ©2019 Men et al.
Copyright year: 2019
Copyright holder: Men et al.
License: This is an open access article distributed under the terms of the Creative Commons Attribution License, which permits unrestricted use, distribution, reproduction and adaptation in any medium and for any purpose provided that it is properly attributed. For attribution, the original author(s), title, publication source (PeerJ Computer Science) and either DOI or URL of the article must be cited.
License URL: https://creativecommons.org/licenses/by/4.0/

Keywords: Collaborative cretivity, Virtual reality, Shared Virtual Environment, Collaborative music making, Sonic interaction design, Gesture design, HCI

Funding: EPSRC and AHRC Centre for Doctoral Training in Media and Arts Technology EP/L01632X/1 China Scholarship Council This work was supported by EPSRC and AHRC Centre for Doctoral Training in Media and Arts Technology (EP/L01632X/1) and China Scholarship Council. The funders had no role in study design, data collection and analysis, decision to publish, or preparation of the manuscript.

==============================
Shared virtual environments (SVEs) have been researched extensively within the fields of education, entertainment, work, and training, yet there has been limited research on the creative and collaborative aspects of interactivity in SVEs. The important role that creativity and collaboration play in human society raises the question of the way that virtual working spaces might be designed to support collaborative creativity in SVEs. In this paper, we outline an SVE named LeMo, which allows two people to collaboratively create a short loop of music together. Then we present a study of LeMo, in which 52 users composed music in pairs using four different virtual working space configurations. Key findings indicated by results include: (i) Providing personal space is an effective way to support collaborative creativity in SVEs, (ii) personal spaces with a fluid light-weight boundary could provide enough support, worked better and was preferable to ones with rigid boundaries and (iii) a configuration that provides a movable personal space was preferred to one that provided no mobility. Following these findings, five corresponding design implications for shared virtual environments focusing on supporting collaborative creativity are given and conclusions are made.

Introduction

The real world envelops us with space that we share with others; in this surrounding environment, we perceive rich sensory information about objects and events happening around us. Using this information, we interact with this outer world around us via inference, manipulation, and exploration. In a similar fashion we interact with other people. In other words, space can be seen as a material of human activity (Raffestin, 2012), and it has a great influence on social activity, e.g., the size of space limits what kind of actions can be performed, the fill material of a space limits how far people can see or hear, and the proxemics between bodies and objects in a space limits their scope of influence.

Digital virtual spaces have existed in different forms for several decades. One of the earliest examples are digital games, e.g., Star Trek created in early 1970s provides a computational space that players can visit and experience through text descriptions on a computer screen, see (Case, Ploog & Fantino, 1990). Though these non-immersive media can involve people to a very high level and generate the experience of flow, few of them have enabled people to interact in a natural way that is similar to the way that people experience real-world interactions. For instance, the interactions based on keyboards and mouse for input and monitor for output have very different properties to real-world interactions (Gaver, 1992). In contrast, virtual reality (VR) provides a novel space for multisensory experiences (Turchet et al., 2018), and enables people to see, hear, and even interact with a virtual space naturally. It offers the potential for people to coordinate collaborative activities in a much more similar way to the real world, presenting people an opportunity to collaborate in virtual space in a more natural way in comparison to non-immersive digital media.

Whilst VR has become a hot topic and has been researched intensively, little attention has been paid to human-human interactions in shared virtual environments (SVEs), with even less being paid to addressing the creative and collaborative aspects of these interactions. We believe that having a better understanding of the role of space and territory within creative collaborations would provide a strong starting point, since real-world collaborations make use of space (Raffestin, 2012) and the demarcation of personal and shared territory is a spatial strategy to affect, influence and control resources and access (Sack, 1983). Hence an effective arrangement and utilisation of a working space can possibly be a crucial factor to a successful collaboration in SVEs too. Thus we are keen on designing and testing working space configurations to see if we can provide more fluid support to creative collaboration in SVEs. We will begin by reviewing related work in SVEs, space, territory, and territoriality. Then the design of our SVE system will be detailed and the study and results will be presented. Finally, the overall study will be discussed and design implications will be given.

Related Work

Shared virtual environments

The term virtual environment (VE) can be traced back to the early 1990s (Bishop & Fuchs, 1992), it emerged as a competing term to virtual reality; however, both are usually equally used to refer to a world created totally by computer simulation (Luciani, 2007). In the mid-1990s, the development of network technology had made it feasible to link many users simultaneously in the same virtual environment, prompting the shared virtual environments (SVEs; Schroeder, 2002). Besides “SVEs”, other terms used include multi-user virtual environments, multi-user virtual reality (Carlsson & Hagsand, 1993), collaborative virtual environments (CVEs; Zhang & Furnas 2003) and social virtual reality (SVR). To align with mainstream usage, we will herein use the term SVEs to refer to VE systems in which users experience other participants as being mutually present in the same environment and in which they can interact inter-personally (cf. Schroeder, 2002). SVEs can be seen as a convergence of research interests in VR and Computing Supported Cooperative Work (CSCW; Benford et al. 2001).

Whilst single-person VEs may focus on creating a detailed (visual) simulation, the design of SVEs typically prioritises enabling collaboration between users (Nassiri, Powell & Moore, 2010). By enabling multiple people to communicate and interact with each other and providing a natural medium for three-dimensional CSCW (Billinghurst et al., 2000), SVEs are considered emerging tools for a variety of purposes, including community activities (Lea, Honda & Matsuda, 1997), online education (Roussos et al., 1997), distributed work and training (Nedel et al., 2016), and gaming and entertainment (see Toybox1 and a PlayStation VR demo2) Despite this, little research exist in the field of supporting collaborative creativity, leaving an open question: how should shared virtual environments be designed to support creative collaboration (cf. Basdogan et al., 2000).

Space, territory, and territoriality

SVEs constitute virtual spaces, although illusive they are meaningful (Steuer, 1992). We believe gaining a better understanding of the virtual space is an effective way to answer the aforementioned question. “Space” is a material given prior to the happening of actions, and territory emerges as a result of the actions and a production of the actors (Raffestin, 2012), helping people mediate their social interaction (Altman, 1975), which is argued to be a key element to collaboration (Kreijns, Kirschner & Jochems, 2003). Additionally, people were found to perform creative collaboration in a similar way with the real world, they divided the working space and formed territory (Men et al., 2017; Men & Bryan-Kinns, 2019). Thus, potentially, with more knowledge of the virtual space, we can even manipulate the virtual space to influence the collaboration in SVEs. Note in this paper, the term “space” specifically refers to the physical virtual space rather than the space concept in psychology or social science, which falls out of the scope of this paper.

Personal and group space in collaboration

A “personal space” herein refers to a specific space assigned to a specific person and “group space” refers to a specific space assigned to a specific group prior to the start of activities (e.g., an experiment). In CSCWs that focus on supporting collaborative creativity, providing personal space is argued to be useful (Fencott & Bryan-Kinns, 2010; Men & Bryan-Kinns, 2019), and integrating personal and group spaces, allowing users to work individually in their personal spaces at their own pace, cooperatively work together in the shared space, and smoothly transition between both is important (Greenberg, Boyle & LaBerge, 1999; Sugimoto, Hosoi & Hashizume, 2004). As a starting point of this exploration, Greenberg, Boyle & LaBerge (1999) developed a PDA-based prototype named SharedNotes. They observed how users shifted between the two spaces and recommended against a rigid notion of “personal”, instead they suggested the boundary between personal and public should be provided with gradations in subtle and lightweight ways, supporting a fluid transition between personal and public. Following that, Shen, Everitt & Ryall (2003) addressed this concern in their project UbiTable by providing a flexible gradient of sharing semantics. Specifically, rather than the binary notion of public and private space, UbiTable provides an additional semi-private space, in which data is visible but not electronically accessible for others. However, both sets of research were carried out based on 2D media (PDA and projector respectively), which made their findings less informative for VR. Moreover, neither of these two explored more gradations between public space and personal space, i.e., the transition between the public space and personal space is step-less and smoother. In this study, we want to design virtual spatial configurations that provide a more gradual boundary between personal and public spaces and see if this can enable a fluid shift and provide better support for collaboration in VR.

Territory and territoriality in collaboration (SVEs and Tabletop)

In a previous study, we found collaborators formed both personal and group territory during collaborative music making in an SVE, and they also had territorial behaviour, e.g., most musical edits were done inside personal territories (Men & Bryan-Kinns, 2019). By manipulating the virtual spatial configurations of an SVE, the formation of territory and territorial behaviour can be influenced and, and as a result, the collaboration can be influenced (Men & Bryan-Kinns, 2019). Because there is limited research on territoriality in VR, and rich research on this in tabletop-based collaboration, we review territoriality in tabletop research as a supplement, which might be informative as it is also a computer-mediated collaboration that evolves territory. The term “tabletop” here refers to interactive tabletop displays, these usually include high quality projectors, flat panel or plasma displays, and touch-sensitive surfaces (Kruger et al., 2004). These electronic tabletops inherit the collaborative benefits of tables, which greatly compensate computers’ disadvantages in this regard (Scott, Carpendale & Inkpen, 2004). Similarly, territoriality also plays an important role in the tabletop collaboration. Collaborators were found to use different types of territory to serve different needs, including sharing, exchanging or storing working tools and resources (Scott, Carpendale & Inkpen, 2004), though some researchers note that removing territorial constraints can promote exploratory group activity (Xambó et al., 2013). Two main types of territory have been identified from research on screen and tabletop mediated collaboration:

(1) Personal territory for performing independent activities. When provided with a personal territory, users prefer to test their contribution before introducing it to the group work (Fencott & Bryan-Kinns, 2010). This type of territory serves as a safe place to try and develop alternate ideas before publishing the ideas (Tang, 1991). Users have been found to prefer to rotate items toward themselves in the personal territory (Tang, 1991) and perform very few actions in their collaborators’ personal territories (Scott, Carpendale & Inkpen, 2004).

(2) Group territory for performing the main task. In group territory, people create and develop new solutions, transfer resources and provide help (Scott, Carpendale & Inkpen, 2004). It is interesting to note that the orientation properties of objects in the group territory can be used to convey support, to separate ideas or to group products (Tang, 1991).

In terms of designing for territoriality, Scott, Carpendale & Inkpen (2004) proposed four guidelines for designing digital tabletop workspaces: (i) visibility of action; (ii) an appropriate size of workspace; (iii) providing functionality in the appropriate locality; (iv) allowing for the grouping of items to facilitate storage. Furthermore, the visibility and transparency of actions have been found to be important in designing group workspaces, as they help collaborators to monitor each others’ actions, maintaining workspace awareness during collaboration (Pinelle, Gutwin & Greenberg, 2003; Fencott & Bryan-Kinns, 2010). However, this can result in overloaded cognitive information, which some people found to be difficult to handle (Fencott & Bryan-Kinns, 2010). To date, little research has explored how such features of workspace might be designed for collaboration in SVEs.

Experiment Design

Creativity domain: why collaborative music making

Music making, as a collaborative activity that relies on shared goals, understanding and good interpersonal communication, has long been a key form of collaborative creativity (cf. Bryan-Kinns & Hamilton, 2012; Titon & Slobin, 1996). Its unique features make it an excellent activity through which to study collaborative activity. In 2003, Blaine & Fels explored the design criteria of collaborative music-making (CMM) systems and pointed out key features including the media used, player interaction, the systems’ learning curves, physical interfaces and so on. In the same year, with inspiration from Rodden’s (1991) Classification Space for collaborative software, otherwise known as groupware, Barbosa (2003) developed the Networked Music Systems Classification Space, which classifies CMM systems (CMMs) in terms of the time dimension (synchronous/asynchronous) and space dimension (remote/co-located). For instance, Daisyphone (Bryan-Kinns, 2004), which provides shared editing of short musical loops falls into the remote synchronous network music systems in this Classification Space. Other examples include reacTable (Xambó et al., 2013) and BilliArT (Bressan, Vets & Leman, 2017), both of which provide co-located music-making experience, and Ocarina (Wang, 2009), which provides a distributed experience. However, we should note that, despite decades of research into CMMs and SVEs, relatively few SVEs that support CMM have been made. As a result, many basic but crucial questions in this field are waiting to be answered, e.g., how shared virtual environments should be designed to support creative collaboration, such as CMM.

Acoustic attenuation

Sound attenuates as a result of diminishing intensity when travelling through a medium. This feature of sounds enables humans to use their innate spatial abilities to retrieve and localise information and to aid performance (cf. Billinghurst & Kato 2002). Whilst it is hard to adjust the acoustic attenuation of a real medium (e.g., the air) to enhance its potential, within VR, as the audio is simulated, we can simulate an augmented spatialised sound purposely. Research has been done on investigating the impacts of spatialised sounds on user experience in VR, e.g., (Hendrix & Barfield, 1996). However, little research explores how the spatialisation of sound may affect or aid collaboration in SVEs (e.g., CMM). Considering sound is both the primary medium and the final output of the creative task CMM (Men & Bryan-Kinns, 2019), by affecting the audio, different settings of acoustic attenuation can possibly affect the collaboration differently. With the ability to modify the simulated acoustic attenuation in an immersive virtual environment, we can possibly create sonic privacy by augmenting acoustic attenuation, this privacy may then be used as personal space supporting individual creativity in CMM.

LeMo—An SVE for collaborative music making

We created Let’s Move (LeMo3), which enables two users to manipulate virtual music interfaces together in an SVE to create a 16-beat music loop, see Fig. 1. LeMo used in this study and Men & Bryan-Kinns (2019) is an extensively modified version of the previous version (Men & Bryan-Kinns, 2018). Major differences include a new visual presentation of the interface, the ability to add, delete, and re-position music loops, with more freedom in terms of instruments, tempo, volume and pitch. These changes are made to improve the experience of music making and, more importantly, to enable users to re-position and share music loops, which is essential for this study. Hereafter, “LeMo” specifically refers to this modified version. LeMo was programmed in Unity, models and textures were made in Cinema 4D and Photoshop respectively. The run-time environment includes two HTC Vive headsets (each with a Leap Motion mounted, see Fig. 1C). The position and rotation of headsets are tracked by two tracking cameras set around the scene, and hand gestures are tracked by the mounted Leap Motion. Two PCs are used to run LeMo, these are connected and synchronised via a LAN cable.

Figure 1 Participant 4A and 4B are creating music together.

LeMo has three key elements: (1) Music interface—LeMo allows users to generate, remove, position and edit virtual music interfaces, which have two modes: sphere and matrix (Fig. 2B). Users can generate up to eight spheres with pinch and stretch gesture, see Fig. 2A. The music interface can be switched between the two modes, re-positioned or removed by manipulating the sphere or the pop button of the matrix with corresponding gestures. As shown in Fig. 3, the matrix interface contains a grid of 16 × 8 dots, with controllers at the bottom. Each row represents the same pitch, forming an octave from bottom to top. Users can edit notes by tapping dots. A vertical play-line repeatedly moves from left to right playing corresponding notes. In this way, each interface generates a 16-note music loop. Three controllers (tempo, volume and pitch) and two functional buttons (erase and switch) are located at the bottom of the matrix interface. (2) Avatars—Each user has an avatar, including a head and both hands (Fig. 1). Avatars are synchronised with users’ real movements in real time, including position and rotation of heads, as well as gestures. LeMo provides visual aids for collaboration by synchronising the virtual environment (virtual space and music interfaces) and avatars across a network, providing participants with the sense of being in the same virtual environment and manipulating the same set of interfaces. (3) A virtual space that includes a grey stage with a grid pattern (part of it is shown in Fig. 1D). Four types of spatial configurations were designed for this study, which will be detailed later.

Figure 2 (A) The gesture to generate a new interface; (B) Matrix (opened interface) and sphere (packed interface), double click the pop button to switch in between (reproduced from Men & Bryan-Kinns 2019).

Figure 3 A C major scale, starting from C4 and finishing at C5, and going back to C4.

Beside these three fundamental elements, LeMo also has: Spatialised audio (volume drops with distance) so that users can hear where the sounds originate; A voice notification system to facilitate the experiments, e.g., in experimental scenario users will hear “1 min left” and “end of session” notifications; A data-log system based on trackers (HTC Vives and Leap Motion) to log time-stamped data from events generated by users’ interactions and movements, including positions and rotations of heads, index fingers, thumb fingers, the manipulation with musical interface (addition/deletion/re-positioning of musical interfaces, addition and deletion of music notes), usage of personal space (activation/deactivation of personal spaces).

Hypotheses

Research has suggested that users should be allowed to work individually in their personal spaces at their own pace, cooperatively work together in the shared space, and smoothly transition between both of the spaces during collaboration (Greenberg, Boyle & LaBerge, 1999; Shen, Everitt & Ryall, 2003; Sugimoto, Hosoi & Hashizume, 2004). In a previous study (Men & Bryan-Kinns, 2019), following this implication, we built three different spatial configurations (public space only, public space + publicly visible personal space, public space + publicly invisible personal space), and tested different impacts of these spatial configurations on collaborative music making in an SVE. The results showed adding personal space to be helpful in supporting collaborative music making in SVE, since it provided a chance to explore individual ideas, and provided higher efficiency. However, several negative impacts also showed up along with the addition of personal space, e.g., longer average distance between participants, reduced group territory and group edits (Men & Bryan-Kinns, 2019). We believe this might due to: (i) the separated stationary locations of the personal spaces, forced collaborators to leave each other to access, and resulted in a longer distance between collaborators and less collaboration; (ii) the rigid boundary between public space and personal space made users more isolated, resulting in a higher sense of isolation. Thus, we are keen on designing new types of personal territory to eliminate these disadvantages, and to provide a more flexible, more fluid collaboration experience. To increase the flexibility, we want to enable users to use personal space anywhere on the stage in the SVE, and see how this flexibility might positively affect the collaboration, thus H1 was developed.

To make the shift between personal and public spaces more fluid, inspired by the implication that the separation between public and personal workspace should be gradual rather than too rigid (Greenberg, Boyle & LaBerge, 1999), we think the attenuation feature can be applied to form a gradual personal space and enable a fluid transition between personal space and public space. This is because the sound is both the primary medium of collaborative tasks and the final work of CMM (Men & Bryan-Kinns, 2018), by manipulating acoustic attenuation, we can produce sonic privacy. For example, different levels of attenuation can lead to different levels of sonic privacy, and a high level of sonic privacy may play a similar role of personal space, thus H2 was developed. Additionally, the acoustic attenuation, rather than a personal space with rigid separation from public space, enables a gradual shift between personal and public workspace, which may possibly increase the fluidity of the experience and support collaboration better (cf. Greenberg, Boyle & LaBerge 1999). Thus we developed H3. Below are the three hypotheses:

H1—Personal space with mobility provides better support for collaboration than personal space with no mobility.

H2—Attenuation can play a similar role to personal space with rigid form (cf. Men & Bryan-Kinns 2019) in CMM in SVE, providing collaborators with a personal space and supporting individual creativity during the collaboration.

H3—Acoustic attenuation provides a fluid transition (no hard borders nor rigid forms) between personal and public spaces, which supports collaboration better compared to conditions with rigid borders.

Independent variable

Spatial configuration is the independent variable in this experiment. As shown in Fig. 4, four space configurations were designed as the independent variable levels to investigate these three hypotheses, including:

Figure 4 Components of the virtual experiment scene. Note: the rigidly soundproof personal space and the handle switch were only available in Cfix & Cmov (A); Tilted view of the four experimental condition settings (B, C, D, E).

Condition 1: Public space only (referred to as Cpub): where players can generate, remove or manipulate music interfaces, and have equal access to all of the space and the music interfaces. As no personal space is provided, a shift between public and personal space does not exist, and users cannot shift to personal space.

Condition 2: Public space + Augmented Attenuation Personal Space (referred to as Caug). In addition to Cpub, the acoustic attenuation is augmented. The volume of audio drops much faster, creating a sonic privacy, which can be seen as a personal space. As the volume changes gradually with the changes in distance, the shift between personal space and public space is gradual.

Condition 3: Public space + Fixed Personal Space (referred to as Cfix). In addition to Cpub, each user is now provided with a personal space located at the corner of the stage (see Fig. 4), which works like a acoustically solid (soundproof) boundary between public space and personal space. In other words, the shift between personal space and public space is now rigid. Each user has a handle to activate/deactivate the personal space, the handle appears automatically over their head when they look up.

Condition 4: Public space + Moveable Personal Space (referred to as Cmov). Every feature of this condition is the same as Cfix, except now the personal space appears centring the user’s current head’s position when being triggered.

Note: to make conditions more realistic and less artificial, the acoustic attenuation in Cpub, Cfix, Cmov are set to mimic the real acoustic attenuation in the real world rather than no attenuation at all.

Dependent variables

To identify how users use the space and the effect of adding personal space with different properties, a series of dependent variables were developed, which can be split into Participant Reports and Activity Assessment.

Participant reports

Questionnaires were used to collect participants’ subjective assessment of the conditions and their experience of the collaboration. The Igroup Presence Questionnaire (IPQ) was used to inform the design of questions about the sense of the collaborator’s presence (Schubert, Friedmann & Regenbrecht, 2001). Questions about output quality, communication, and contribution were adapted from the Mutual Engagement Questionnaire (MEQ; Bryan-Kinns, 2013). The rest of the questions were designed to question people’s preference for conditions. The questionnaire included questions on:

(1) Presence: (i) sense of co-worker’s presence and (ii) sense of collaborator’s activities. Note that the measure “sense of self-presence” is not included in this study, because a previous study (Men & Bryan-Kinns, 2019) has shown LeMo’s capability on producing a high-level of self-presence.

(2) Communication: quality of communication, which may vary as the visibility of spaces can possibly affect the embodiment and nonverbal communication.

(3) Content assessment: the satisfaction of the final music created reflects the quality of collaboration, cf. (Bryan-Kinns, 2013; Bryan-Kinns & Hamilton, 2012).

(4) Preference: preference of the conditions, checking if users have subjective preferences towards the settings.

(5) Contribution: (i) the feeling of self’s contribution; (ii) the feeling of others’ contribution. These measures were set to see the effects of spatial configurations on the sense of contribution.

These measures will be grouped into a Post-Session Questionnaire (PSQ, see its items in Table 1), to be filled after participants experiencing each condition, and a Comparison Questionnaire (CQ, see its items in Table 2), to be filled at the end of the experiment.

Activity assessments

To access the characteristics of collaboration, we developed the following measures of activity in the collaboration based on the system-logged data:

(1) Contribution: (i) number of musical note edits; (ii) number of note additions/deletions; (iii) number of mutual note modifications. Here a mutual note modification indicate an edit on a certain note, the last update of which was performed by the collaborator, cf. (Bryan-Kinns, Healey & Leach, 2007).

(2) Time and amount of use of personal space (only in condition Cfix and Caug, as this measure is based on the time-stamped entries of the rigid boundary of personal spaces): (i) number of uses of, (ii) length of time of using, and (iii) average duration of each use of personal space.

(3) Location and territory: (i) distribution of participants’ locations and interactions; (ii) the sizes of personal/group territory; (iii) note edits fallen in different types of territory; (iv) average distance between participants, cf. colocation in (Bryan-Kinns, 2013).

(4) Attention: (i) time participants spent paying close/ordinary attention to collaborator; (ii) number of times paying close/ordinary attention to the collaborator. Strictly speaking, here “paying attention” means “facing toward the collaborator‘s avatar” as no eye tracker was involved in this study.

Table 1 Results of Post-Session Questionnairea and results of Wilcoxon Rank Sum Test (two-tailed)b.

Questions	Cpub	Caug	Cfix	Cmov	CpubvsCaug	CpubvsCfix	CpubvsCmov	CaugvsCfix	CaugvsCmov	CfixvsCmov	
	M (SD)	M (SD)	M (SD)	M (SD)	p (W)	p (W)	p (W)	p (W)	p (W)	p (W)	
PSQ1 (support for creativity)—I think the spatial configuration in this session was extremely helpful for creativity	
	8.55
(1.44)	8.77
(1.34)	7.61
(2.01)	7.82
(1.94)	0.5695
(259)	0.07706
(396.5)	0.1563
(379)	0.02372
(492)	0.06318
(469)	0.695
(368)	
PSQ2 (support for creativity)—I feel like the spatial configuration in this session was extremely helpful to support the development of my own ideas	
	7.82
(1.92)	8.35
(1.50)	7.71
(1.88)	7.75
(1.62)	0.5211
(255.5)	0.6456
(331.5)	0.5029
(342)	0.2172
(434)	0.1452
(446.5)	0.8999
(400)	
PSQ3 (preference)—I enjoyed the spatial configuration of this virtual world very much	
	8.27
(1.61)	8.65
(1.60)	8.18
(1.87)	8.07
(1.86)	0.2622
(233)	0.9358
(303.5)	0.3863
(311.5)	0.3863
(412.5)	0.2165
(433.5)	0.8010
(407.5)	
PSQ4 (sense of collaborator’s presence)—I always had a strong feeling that my collaborator was there, collaborating with me together, all the time	
	8.91
(0.92)	8.54
(1.68)	7.07
(2.52)	7.93
(2.26)	0.7961
(298.5 )	0.004813
(450)	0.1636
(377.5)	0.01946
(497)	0.3229
(420)	0.1368
(302)	
PSQ5 (content assessment)—How satisfied are you with the final piece of loop music you two created in this session	
	8.64
(1.73)	8.38
(1.50)	7.21
(2.22)	8.32
(1.96)	0.4287
(323.5)	0.005155
(448.5)	0.5557
(337.5)	0.05449
(473.5)	0.803
(349.5)	0.02163
(254)	
PSQ6 (communication quality)—How would you rate the quality of communication between you and your collaborator during the session	
	8.68
(1.09)	8.50
(1.36)	7.04
(2.25)	8.04
(1.97)	0.7644
(300.5)	0.004494
(450.5)	0.3038
(359)	0.01038
(510)	0.5404
(399)	0.05073
(274)	
PSQ7 (sense of collaborator’s activity)—I had a clear sense of what my collaborator was doing	
	8.73
(1.20)	7.96
(1.54)	6.50
(2.52)	7.29
(2.49)	0.08094
(368.5)	0.0003856
(487.5)	0.03436
(414.5)	0.02786
(489.5 )	0.5095
(402)	0.176
(310)	
PSQ8 (amount of contribution)—The amount of your contribution to the joint piece of music is	
	8.41
(1.44)	8.15
(1.46)	6.96
(2.15)	7.50
(1.67)	0.4776
(320)	0.009236
(439.5)	0.03928
(412)	0.04281
(479.5)	0.166
(443)	0.4489
(346)	
PSQ9 (amount of contribution)—The amount of your collaborator’s contribution to the joint piece of music is	
	8.18
(1.26)	8.23
(1.39)	7.29
(1.96)	7.61
(1.97)	0.8486
(276.5)	0.08916
(394)	0.4025
(350.5)	0.06406
(469.5)	0.3008
(423)	0.4739
(348.5)	
PSQ10 (quality of contribution)—What do you think of the quality of your contribution to the joint piece of music is	
	8.05
(1.70)	7.81
(1.41)	7.36
(1.68)	7.86
(1.53)	0.319
(333.5)	0.1031
(390)	0.4648
(345)	0.3596
(416.5)	0.2829
(327)	0.8599
(353.5)	
PSQ11 (quality of contribution)—What do you think of the quality of your collaborator’s contribution to the joint piece of music is	
	7.73
(1.52)	8.19
(1.20)	7.54
(1.50)	7.75
(2.05)	0.3496
(241.5)	0.5636
(337.5)	0.6459
(284.5)	0.1143
(453.5)	0.6992
(386)	0.3559
(336.5)	
Notes.

a With 10-point-Likert scale, 1 indicates no fulfilment at all with the description of the questionnaire and 10 indicates a full fulfilment.

b Note statistics in this table are calculated based on the data collected from the third and fourth session to counterbalance the learning effect.

Participants and procedure

Fifty-two participants (26 pairs) were recruited for this study4 via emails sent to group lists within the authors’ school. All participants were aged between 18 and 35, with an average age of 23.00 (SD = 4.37). Participants’ mean rating of personal musical theory knowledge is 3.92 (SD = 2.50) on a 10-point Likert scale, where higher values indicate increased knowledge. Twenty four (46.15%) played one or more instruments, and the remaining 28 (53.85%) did not. Participants’ rating of experience of collaboratively composing music is 2.13 (SD = 1.56) on a 10-point Likert scale, where higher values indicate increased experience. Regarding familiarity with computers, 31 (59.62%) participants rated themselves as computer “experts”, 20 (38.46%) chose “intermidiate”, and only 1 participant (0.02%) chose “beginner”. Twenty participants (38.46%) had tried VR 2–5 times before, 20 (38.46%) had only tried once, the remaining 12 (23.08%) had no VR experience before. Thirty-seven participants knew their collaborators very well prior to the experiment, 3 met their collaborators several times, but did not know well, the remaining 12 did not know their collaborators at all prior to the experiment.

Table 2 Results of Binomial Test of Comparison Questionnaire (CQ)a.

Question description	Option	Cpub	Caug	Cfix	Cmov	
		k	p	k	p	k	p	k	p	
CQ1 (preference)—In which session, you enjoyed the spatial configuration the most?	
	most enjoyed	10	0.2146	16	0.2089	10	0.2146	16	0.2089	
	least enjoyed	15	0.3084	10	0.2146	20	0.02205	7	0.03317	
CQ2 (content assessment)—In which session, you made the music you were most satisfied with?	
	most satisfied	16	0.2089	12	0.4469	10	0.2146	14	0.4262	
	least satisfied	13	1.000	9	0.1292	21	0.01054	9	0.1292	
CQ3 (coordination)—Which session you found most difficult to track collaborator’s activities?	
	most difficult	7	0.03317	12	0.4469	20	0.02205	13	1.000	
	least difficult	22	0.004691	14	0.4262	8	0.06971	8	0.06971	
CQ4 (sense of collaborator’s presence)—Which session did you have the strongest sense that your collaborator was there	
   working with you together	
	strongest	27	2.807e−05	17	0.1322	2	5.277e−05	6	0.01368	
	least strongest	4	0.001378	7	0.03317	28	8.12e−06	13	1.000	
CQ5 (communication quality)—Which session did you have the best quality of communication between your self and your collaborator	
	best quality	20	0.02205	17	0.1322	4	0.001378	11	0.3232	
	worst quality	6	0.01368	13	1.000	25	0.0002698	8	0.06971	
CQ6 (preference)—Which session had the best setting for creating a good piece of music collaboratively	
	best setting	16	0.2089	16	0.2089	8	0.06971	12	0.4469	
	worst setting	13	1.000	10	0.2146	19	0.04298	10	0.2146	
CQ7 (coordination)—Which session did you find most difficult to cooperate with collaborator	
	most difficult	7	0.03317	12	0.4469	22	0.004691	11	0.3232	
	least difficult	21	0.01054	14	0.4262	7	0.03317	10	0.2146	
CQ8 (contribution)—Which session do you you feel you made the most contribution to the joint piece	
	most contribution	14	0.4262	12	0.4469	13	1.000	13	1.000	
	least contribution	11	0.3232	13	1.000	13	1.000	15	0.3084	
CQ9 (contribution)—Which session do you you feel your collaborator made the most contribution to the joint piece	
	most contribution	11	0.3232	11	0.3232	16	0.2089	14	0.4262	
	least contribution	15	0.3084	12	0.4469	18	0.07806	7	0.03317	
Notes.

a Lower-tailed test when k < 13, two-tailed test when k = 13, upper-tailed test when k > 13.

The experimental hardware setup was exactly the same with LeMo standard setup, see more details in the previous section “LeMo—An SVE for collaborative music making”. After reading information forms and signing consent forms, each pair of participants first received an explanation of the music interface of LeMo (see Fig. 3). Then one experimenter demonstrated all of the interaction gestures supported in LeMo. By linking the demonstration with the first-person view shown on monitors, participants had a chance to learn how to play LeMo. Then, participants had a trial (5–15 min) to try all the ways of interaction. The trial ended once they were confident enough of all available gestures. The length of time of the tutorial session was flexible to ensure participants with diverse musical knowledge could grasp LeMo. Participants were then asked to have four sessions of collaboratively composing music that was mutually satisfying and complimented an animation loop, each session lasts 7 min. Based on our pilot study and a previous study (Men & Bryan-Kinns, 2018), we found 7 min was sufficient for the task. To counter-balance the learning effect, all four conditions were experienced in a fully randomised sequence to counterbalance the learning effect. In total four animation loops were introduced to trigger participants’ creativity, each to be played in one experimental session on four virtual screens surrounding the virtual stage. These clips were played in an independently randomised sequence to counterbalance impacts on the study. Each session ended with a Post-Session Questionnaire (PSQ, see Table 1). After all the four sessions finished, the Comparison Questionnaire (CQ, see Table 2) and a short interview were carried out at the end of the experiment.

Results

Participant reports

This section reports on the results of the questionnaires. Ratings of Post-Session Questionnaire were refined to counterbalance the learning effect and then analysed with Wilcoxon Rank Sum Tests (Table 1). Binomial tests were run to see if the number of ratings for each option was significantly different than would be expected by chance, upper-tail, lower-tail or two-tailed tests were used accordingly, results are listed in Table 2. Next, we will present how we counterbalanced the learning effect on PSQ and then results will be reported following the sub-type of measures.

Figure 5 Results of Post-Session Questionnaire (N = 52) for all sessions.

Arcs show significant (solid line) and marginal-significant (dotted line) differences between conditions, indicating possible ordering effects.

Counterbalancing the learning effect

As aforementioned, we introduced a fully randomised order of experimental conditions to counterbalance the learning effect. However, it turned out many measurements in the Post-Session Questionnaire were still affected by the sequence to a certain extent, as shown in Fig. 5, in which data from all groups were compiled according to how the group was ordered in the session sequence. Wilcoxon Rank Sum tests were run between each two conditions for every question. A red solid arc indicates a significant difference between two bars (p < 0.05), and a grey dotted arc indicates a trend toward a significant difference (p < 0.1). The arcs show that results of four questions (especially PSQ1, PSQ5, PSQ8, PSQ9, PSQ10, PSQ11) are very sensitive to the sequential position of the session. Specifically, in later sessions, participants responded more positively to the helpfulness of the spatial configuration (PSQ1), higher satisfaction with their output (PSQ5), and both more, and better quality of contributions by themselves and contributors (PSQ8, PSQ9, PSQ10, PSQ11). This is probably due to the learning effect which has a much stronger effect on these measures compared with the differences between experimental conditions. Given the limited experience some participants had in VR and collaborative music making, learning effect could have possibly and greatly promoted participants’ skills and knowledge in performing the task, resulting in a better feeling of the spatial configuration of the session, higher quality of output, more contribution with better quality in later sessions. This learning effect has been also mentioned by some participants in the interview. More details will be discussed in the later subsection “Interviews”.

To better counterbalance the learning effect and habituation on PSQ, only data collected via PSQ in later two sessions (session 3 and 4) will be retained at the expense of the halved sample size. Box-plots were then drawn (Fig. 6) and Wilcoxon Rank Sum tests were run (Table 1) to compare the conditions against each other.

Figure 6 Results of Post-Session Questionnaire (N = 26), data grouped by experimental conditions.

Only data collected in the latter two sessions are included. Solid arcs show significant differences and dotted arcs show marginal-significant differences between conditions.

General feeling (helpfulness of spatial configuration, difficulty of cooperation)

When asked the condition’s helpfulness for creativity (PSQ1 in Table 1), on a 10-point Likert Scale, participants gave an average rating of 8.77 in Caug, which is significantly higher than 7.61 given in Cfix (Wilcoxon Rank Sum Test, W = 492, p = 0.02372). There are trends towards significances between participants’ rating of Caug and Cmov (W = 469, p = 0.06318), and between Cpub and Cfix (W = 396.5, p = 0.07706). These differences indicate that Caug is better than Cfix, and possibly also better than Cmov in terms of supporting participants’ creativity.

When asked to rate the helpfulness of spatial configurations to support personal idea development (PSQ2), the mean rating of Caug (M = 8.35) is higher than that of the other three conditions (Cpub: M = 7.82; Cfix: M = 7.71; Cmov: M = 7.75), though no significant differences were found. CQ7 of Table 2 shows that Cpub was rated by significantly many participants (21 out of 52) to be the least difficult to cooperate with their collaborator (Binomial Test, 0.40 > 0.25, p = 0.01054, 1-sided), whilst significantly few participants rated Cpub as the most difficult one to do so (Binomial Test, 0.13 < 0.25, p = 0.03317, 1-sided). On the opposite, Cfix was rated by significantly many participants as the most difficult (Binomial Test, 0.42 > 0.25, p = 0.004691, 1-sided), and significantly few participants (7 out of 52) rated it as the least difficult (Binomial Test, 0.13 < 0.25, p = 0.03317, 1-sided).

Preference

When asked the level of enjoyment of the spatial configuration (PSQ3), similar to PSQ2, Caug got a higher rating (M = 8.65). However, no significant difference was revealed. In CQ1 of Table 2, when asked which session has the most enjoyable spatial configuration, out of 52 participants, both Caug and Cmov were chosen by 16 participants, more than those choosing Cpub and Cfix (10 participants each), though no significant difference was found. When asked which session had the least enjoyable spatial configuration, a significant number of participants (20 out of 52) opted Cfix (0.38 > 0.25, p = 0.02205, 1-sided), and significantly few (7 out of 52) opted Cmov (0.13 < 0.25, p = 0.03317, 1-sided). Result of CQ6 in Table 2 indicates that significantly many participants (19 out of 52) believed Cfix was? the worst setting for creating a good piece of music collaboratively. To summarise, the spatial configuration of Cfix is more disfavoured and that of Cmov is less disfavoured.

Sense of co-presence

Results of PSQ4, PSQ7 and CQ4 reveal participants’ sense of collaborators’ presence and activities. Cfix’s ratings in PSQ4 and PSQ7 are significantly lower than Cpub and Caug (Wilcoxon Rank Sum Test, all p < 0.05), indicating Cfix saw a lower sense of the collaborator’s presence and activities. Similarly, when being asked in which session they had the strongest sense of collaborators (CQ4), significantly many participants (27) chose Cpub (0.52 > 0.25, p = 2.807e−05, 1-sided), 17 chose Caug, significantly few chose Cmov (chosen by 6) and Cfix (chosen by 2; Binomial Test, both p < 0.05; more details in CQ4 of Table 2). When questions changed to “least sense of collaborator’s presense”, ratings reversed, significantly many chose Cfix while significantly few chose Cpub and Caug (Binomial Test, all p < 0.05; more details in CQ4 of Table 2). These results indicate that in terms of maintaining the sense of collaborator’s presence, Cpub > Caug > Cmov > Cfix.

Regarding the sense of collaborator’s activities (PSQ7), a significantly weaker sense was reported in Cfix compared with Cpub and Caug (Wilcoxon Rank Sum Test, both p < 0.05). Cpub also saw a stronger sense compared with Cfix (Wilcoxon Rank Sum Test, W = 414.5, p < 0.0346). No significant difference was found between Cpub and Caug nor between Cfix and Cmov. Similarly, CQ3 of the Comparison Questionnaire reveals that significantly many participants reported tracking collaborators’ activities in Cfix was the most difficult, and significantly many felt least difficult to do so in Cpub (Binomial Test, both p < 0.05). These indicate that Cpub seems to be easier for participants to track collaborators’ activities, and Cfix is more difficult for them to do so.

Content assessments

Participants reported a mean rating 7.21 of output quality in Cfix (PSQ5 of Table 1), which is significantly lower than 8.64 in Cpub, and 8.32 in Cmov (Wilcoxon Rank Sum Test, both p < 0.05), and quasi-significantly lower than 8.38 in Caug (W = 473.5, p = 0.05549). Similarly, significantly many participants reported that they produced the least satisfying piece of music in Cfix (Binomial Test, 0.40 > 0.25, p = 0.01054, 1-sided), see CQ2 of Table 2. In other words, Cfix tended to led to a music output with lower quality compared with other sessions.

Communication assessments

As listed in PSQ6, communication quality of Cfix (M = 7.04) is significantly lower than 8.68 of Cpub and 8.50 of Caug (Wilcoxon Rank Sum Test, both p < 0.05), and near-marginal significantly lower than 8.04 of Cmov (Wilcoxon Rank Sum Test: W = 274, 0.05073), see PSQ6 in Table 1 and PSQ6 in Fig. 6. When asked to compare these sessions, significantly many participants reported the best communication quality was in Cpub and significantly few believed they had the best communication quality in Cfix (Binomial Test, both p < 0.05, see CQ5 of Table 2). Conversely, significantly few had the worst communication quality in Cpub and significantly many had worst in Cfix (Binomial Test, both p < 0.05, see CQ5 of Table 2). In summary, Cfix saw a relatively lower communication quality.

Contribution

Participants reported that they did a significantly larger amount of contributions in Cpub compared with Cfix (W = 439.5, p = 0.009236) or with Cmov (W = 412, p = 0.03928), and had done significantly more contribution in Caug compared with Cfix (W = 479.5, p = 0.04281), see PSQ8. No significant difference was found in CQ8, which is also questioning the feeling of own contribution.

No significant differences were found in the ratings of the amount of the collaborators’ contribution (PSQ9), except a trend reporting their collaborator had a lower amount of contribution in Cfix than Cpub and Caug (Wilcoxon Rank Sum Test, W = 469.5, both p < 0.1). However, CQ9 reveals significantly few participants felt that their collaborator did the most contribution in Cmov (Binomial Test, 0.13 < 0.25, p = 0.03317, 1-sided). These results indicate that the addition of personal space in Cfix and Cmov possibly led to a weaker sense of collaborator’s activities.

Activity assessments

This section reports on measures focusing on the participants’ interactive activities. All measures are listed in Table 3, Wilcoxon Rank Sum tests were run to compare conditions against each other.

Table 3 Statistics and Wilcoxon Rank Sum Test (two-tailed) of Activity Assessments (AA).

Measure Cpub	Caug	Cfix	Cmov	CpubvsCaug	CpubvsCfix	CpubvsCmov	CaugvsCfix	CaugvsCmov	CfixvsCmov	
	M (SD)	M (SD)	M (SD)	M (SD)	p (W)	p (W)	p (W)	p (W)	p (W)	p (W)	
AA1—No. of note edits	
	77.13
(36.59)	80.27
(36.92)	98.35
(48.67)	77.69
(34.61)	0.6988
(1292)	0.02386
(1004)	0.7599
(1304.5)	0.03375
(1025)	0.8965
(1372.5)	0.02228
(1704)	
AA2—No. of note additions	
	50.23
(27.12)	58.96
(30.03)	72.88
(40.93)	55.98
(25.31)	0.8301
(1318.5)	0.03429
(1026)	0.8149
(1315.5)	0.06572
(1068.5)	0.876
(1376.5)	0.05591
(1646.5)	
AA3—No. of note deletions	
	20.90
(14.46)	21.31
(12.94)	25.46
(18.39)	21.71
(15.15)	0.7108
(1294.5)	0.243
(1172)	0.8376
(1320)	0.3308
(1202)	0.9689
(1358.5)	0.3323
(1501.5)	
AA4—No. of mutual note modifications a	
	4.37
(4.42)	4.23
(5.57)	3.71
(7.69)	2.44
(3.92)	0.6452
(1422.5)	0.01929
(1703.5)	0.007331
(1754.5)	0.06614
(1627.5)	0.02514
(1687.5)	0.7732
(1394.5)	
AA5—Size of group territory (unit: m2)	
	0.3465
(0.2443)	0.4331
(0.2446)	0.2339
(0.1878)	0.3103
(0.1942)	0.152
(259)	0.1013
(428)	0.7099
(359)	0.005236
(491)	0.09421
(430)	0.2639
(276.5)	
AA6—Size of personal territory (unit: m2)	
	0.4282
(0.1690)	0.4547
(0.2193)	0.7475
(0.1801)	0.5067
(0.1894)	0.9559
(1343)	2.25e−12
(272)	0.02347
(1003)	2.085e−10
(374)	0.04421
(1042)	2.3e−08
(2212)	
AA7—No. of group edits (note edits done in group territory)	
	36.44
(35.24)	43.04
(34.79)	17.50
(23.79)	25.23
(29.00)	0.2913
(1189.5)	0.001448
(1837)	0.07839
(1621.5)	4.043e−05
(1977)	0.009044
(1751.5)	0.1822
(1151)	
AA8—No. of personal edits (note edits done in own personal territory)	
	40.50
(44.81)	37.10
(38.42)	80.62
(51.89)	52.42
(38.81)	0.9610
(1360)	1.179e−05
(678)	0.0294
(1017)	2.157e−06
(623)	0.02016
(994.5)	0.003695
(1799)	
AA9—No. of note edits done in other’s personal territory	
	0.058
(0.42)	0
(0)	0.19
(0.89)	0.019
(0.14)	0.3267
(1378)	0.1797
(1275)	1.000
(1352.5)	0.04343
(1248)	0.3267
(1326)	0.1686
(1431)	
AA10—Average distance between collaborators (unit: metre)	
	1.11
(0.38)	1.12
(0.38)	2.19
(0.58)	1.28
(0.41)	0.8632
(348)	4.731e−11
(26)	0.2119
(269)	2.663e−10
(34)	0.08045
(242)	2.459e−08
(616)	
AA11—No. of uses of personal spaces	
	-
(-)	-
(-)	2.40
(1.95)	2.85
(2.14)	-
(-)	-
(-)	-
(-)	-
(-)	-
(-)	0.2912
(1193)	
AA12—Length of time of using personal spaces (unit: second)	
	-
(-)	-
(-)	128.60
(86.95)	112.19
(78.67)	-
(-)	-
(-)	-
(-)	-
(-)	-
(-)	0.4685
(1464)	
AA13—Average duration of each entry of personal space (unit: second) b	
	-
(-)	-
(-)	73.07
(56.55)	49.54
(44.83)	-
(-)	-
(-)	-
(-)	-
(-)	-
(-)	0.008019
(1512)	
AA14—No. of note edits in public space	
	-
(-)	-
(-)	54.48
(48.69)	43.69
(34.96)	-
(-)	-
(-)	-
(-)	-
(-)	-
(-)	0.5051
(1455)	
AA15—No. of note edits in personal space	
	-
(-)	-
(-)	43.87
(40.10)	34
(25.37)	-
(-)	-
(-)	-
(-)	-
(-)	-
(-)	0.3869
(1485.5)	
AA16—Time spent paying close attention to collaborator (unit: second) c	
	7.19
(7.44)	14.04
(15.19)	5.51
(9.60)	9.43
(13.96)	0.01032
(957)	0.364
(1492)	0.4725
(1241)	0.001757
(1833.5)	0.05722
(1645)	0.1088
(1105)	
AA17—Times of paying close attention to collaborator c	
	9.31
(8.33)	14.79
(11.16)	7.31
(7.48)	11.02
(11.62)	0.005451
(924.5)	0.1591
(1568.5)	0.446
(1234.5)	0.0001145
(1945)	0.03122
(1683.5)	0.02838
(1015)	
AA18—Time spent paying ordinary attention to collaborator (unit: second) c	
	79.38
(58.65)	76.32
(51.55)	52.89
(39.91)	74.68
(60.69)	0.9818
(1356)	0.03719
(1673)	0.6655
(1419)	0.02047
(1709)	0.5695
(1440)	0.07865
(1081)	
AA19—Times of paying ordinary attention to collaborator c	
	45.02
(19.23)	46.77
(19.42)	32.87
(13.55)	44.98
(20.67)	0.7973
(1312)	0.0006613
(1876)	0.8888
(1374)	0.0001704
(1930.5)	0.6773
(1416.5)	0.002744
(891)	
AA20—No. of music interface additions	
	x
(x)	x
(x)	x
(x)	x
(x)	x
(x)	x
(x)	x
(x)	x
(x)	x
(x)	x
(x)	
AA21—No. of music interface additions in public space	
	-
(-)	-
(-)	x
(x)	x
(x)	-
(-)	-
(-)	-
(-)	-
(-)	-
(-)	x
(x)	
AA22—No. of music interface additions in personal space	
	-
(-)	-
(-)	x
(x)	x
(x)	-
(-)	-
(-)	-
(-)	-
(-)	-
(-)	x
(x)	
Notes.

a Mutual note modifications include activation/deactivation, the last update of which was performed by the collaborator.

b Data of four participants (3B, 4A, 17B 18A) were excluded when calculating this metric as these participants did not use personal space, which made this metric not apply to them.

c The difference between the close attention and the ordinary attention is the breadth and depth of FOV, FOV of close attention roughly covers 27 degrees (horizontally), 28 degrees (vertically) and 1 m (depth), whilst FOV of ordinary attention roughly covers 27 degrees (horizontally), 28 degrees (vertically) and 2.7 m (depth).

Contribution

(1) Note edits (including note additions and deletions). On average, participants did 98.35 note edits in Cfix, which is significantly more than 77.13 of Cpub, 80.27 of Caug, and 77.69 of Cmov (Wilcoxon Rank Sum Test, all p < 0.05). Note additions, as the main part of note edits, follow a similar pattern. The number of note additions in Cfix is significantly greater than that of Cpub (W = 1026, p = 0.03429), and near-marginal significantly greater than that of Caug and Cmov (Wilcoxon Rank Sum Test, both p < 0.07, check detailed statistics in AA2, Table 3). No significant difference was found in note deletions between conditions, this is probably due to the much smaller amount of deletions compared with the number of note edits and additions. These results indicate that participants had more musical edits, specifically note additions in Cfix than the other conditions.

(2) Mutual note modifications. Cpub saw the highest average number of mutual note modifications (M = 4.37, SD = 4.42), this is significantly more than Cfix (M = 3.71, SD = 7.69; Wilcoxon Rank Sum Test, W = 1703.5, p = 0.01929) and Cmov (M = 2.44, SD 3.92; Wilcoxon Rank Sum Test, W = 1754.5, p = 0.007331). Caug has the second highest mean (M = 4.23, SD = 5.57), which is significantly more than Cmov (W = 1687.5, p = 0.02514), and near-marginal significantly more than that of Cfix (W = 1068.5, p = 0.06614). No significant difference between Cpub and Caug or between Cfix and Cmov was found. These results indicate participants had more mutual modifications in Cpub and Caug than Cfix and Cmov, which might indicate a closer collaboration.

(3) Number of note edits that fell into public/personal space. Note this measure is only applicable to rigid personal space, which were only available in Cfix and Cmov. Participants did 54.48 (SD = 48.69) note edits in public space, 43.87 (SD = 40.10) note edits inside personal space in Cfix, these numbers reduced to 43.69 (SD = 34.69) in public space and 34 (SD = 25.37) inside personal space when it comes to Cmov. Although both numbers decreased, no significant differences were found between conditions.

Location and territory

To illustrate how participants used the space, based on the system-logged data, we plotted the positions and directions of participants’ heads and musical note edits on a top view of the stage, see Fig. 7 as an illustrative example of visual traces from arbitrarily selected group 3. Visual traces of all groups are shown in Fig. 8. These figures were made based on system logged data, specifically, the arrows were participants’ locations at 20-second intervals for ease of reading the diagram, and dots are the locations of participants’ hands when making musical note edits. Research of table-top collaboration defines personal territory as a workspace close to the person and group territory as the central area or spaces between collaborators (Xambó et al., 2013; Scott, Carpendale & Inkpen, 2004; Scott & Carpendale, 2010). Following this definition, we dye the area within a 0.6-metre radius of the participants’ locations (locations here are at 1-second interval for higher accuracy) with different tint colours (red for participant A’s personal territory, and blue for B’s) to indicate territories. We chose 0.6 metres as it falls into the range of close phase of personal distance, and permits participants to touch each other or the same music interface (Hall, 1966), most of the musical note edits also fell inside this range.

Figure 7 Illustrative example of visual traces of the participants’ locations, directions and musical note edits (group 3).

Arrows show participant’s position and direction at 20 s intervals, dots show participant’s hand’s position while performing note edits.

Figure 8 Visual traces—the participants’ locations, directions and musical note edits shown on a top view of the stage (based on system-logged data of all groups).

(1) Distribution of locations and interactions. The more intensely blue or red the area is, the more presence the corresponding participant had shown in that location. The overlap is coloured grey, indicating the appearance of both participants, which can be seen as group territory.

(2) Sizes of group territory and group edits (edits undertaken in group territory). By calculating the size of red/blue/grey area, the size of personal/group territory can be calculated. Specifically, participants formed an average of 0.3465 m2 of group territory in Cpub, 0.4331 m2 in Caug, 0.2339 m2 in Cfix and 0.3103 m2 in Cmov (AA5 of Table 3). Results of the Wilcoxon Rank Sum Tests show that the size of group territory of Caug is significantly larger than that of Cfix (W = 491, p = 0.005236), and near-marginal significantly larger than Cmov (W = 430, p = 0.09421). No significant difference was found between Cpub and Caug.

AA7 of Table 3 shows that participants had an average of 36.44 group edits in Cpub, which is significantly more than 17.50 of Cfix (Wilcoxon Rank Sum Test, W = 1837, p = 0.001448), and a near-marginal significantly more than 25.23 of Cmov (W = 1621.5, p = 0.07839). Caug resulted in a higher average of group edits (M = 43.04), though not significantly higher than Cpub, it is significantly higher than numbers of Cfix and Cmov (Wilcoxon Rank Sum Test, both p < 0.01). These indicate the spatial configurations of Cpub and Caug are more friendly to group edits.

(3) Sizes of personal territory (AA6 of Table 3) and personal edits (edits fallen into personal territory; AA8). Participants formed a significantly larger personal territory in Cfix (M = 0.7475 m2, SD = 0.1801) compared with all the other three conditions (Wilcoxon Rank Sum Test, all p < 0.001), and had significantly more personal edits in Cfix compared with other conditions (Wilcoxon Rank Sum Test, all p < 0.01; AA8). Similarly, a larger size of personal territory was formed in Cmov and more personal edits were done in Cmov compared with Cpub and Caug (all p < 0.05). No significant differences were found between Cpub and Caug, neither in the size of personal territory nor in the number personal edits. To summarise, Cfix results in the largest size of personal territory and the largest number of personal edits, the metrics of Cmov follows, and Cpub and Caug have the least, indicating that Cfix led to a much looser collaboration, in which participants worked more independently, whilst Cpub and Caug, on the opposite, led to more interactivities in the group territory.

(4) Average distance (AA10 of Table 3). Participants had an average distance of 2.19 metres between themselves and their collaborators in Cfix, this is significantly bigger than other three conditions (Wilcoxon Rank Sum Test, all p < 0.001). Namely, in the other three sessions participants worked more closely.

Times and amount of use of personal space

As shown in AA11, AA12 and AA13 of Table 3, in Cfix, participants had an average of 2.40 entries of personal space on average, with each entry lasting 73.07 s with total duration 128.60 s. For Cmov, the participants did 2.85 entries, each lasting 49.54 s on average, with a total usage time of 112.19 s, no significant difference was found in the number of entries or in the sum usage time. However, the average duration of each entry of Cmov is significantly shorter than that of Cfix (Wilcoxon Rank Sum Test, W = 1512, p = 0.008019), indicating that the personal spaces of Cfix were possibly more used for longer, independent creation.

Attention

(1) The time participants spent paying close attention to each other - Throughout the 420-second session, participants had their close attention toward their collaborators’ heads for 14.04 s in Caug, which is significantly longer than 7.19 s in Cpub, and 7.31 s in Cfix (Wilcoxon Rank Sum Test, both p < 0.05), and near-marginal significantly longer than 11.02 s in Cmov (W = 1645, p = 0.05722, see AA16 in Table 3).

(2) Participants oriented their close attention toward their collaborator for significantly different times, they did most of the time in Caug (M = 14.04), this is significantly more than 7.19 in Cpub, 7.31 in Cfix and 11.02 in Cmov (Wilcoxon Rank Sum Test, Cpub vs Caug: W = 924.5, p = 0.005451; Caug vs fix: W = 1945, p = 0.0001145; Caug vs Cmov: W = 1683.5, p = 0.03122). Wilcoxon Rank Sum Test result also shows participants paid their attention to their partner significantly fewer times in Cfix than they did in Cmov (W = 1015, p = 0.02838), see AA17 in Table 3. These results indicate the spatial configuration in Caug significantly promoted participants to pay more close attention to their collaborator, whilst Cmov possibly promoted insignificantly and Cfix demoted insignificantly compared with Cpub.

(3) The time participants spent paying ordinary attention to each other. Different from the impact of spatial configurations on close attention, neither Caug nor Cmov significantly changed the way participants paying their ordinary attention, all fell inside the range from 70 to 80 s out of the 420-second session. Cfix greatly reduced participant’s ordinary attention paid to each other, on average participants only spent 52.89 s on doing this, which is significantly shorter than Cpub and Caug (Wilcoxon Rank Sum Test, both p < 0.05) and near-marginal significantly shorter than Cmov (Wilcoxon Rank Sum Test, W = 1081, p = 0.07865), more details are shown in AA18 of Table 3.

(4) Similar to the time paying ordinary attention to each other, participants only drew an average of 32.87 times of ordinary attention to each other in Cfix, which is significantly lower than all the other three conditions (Wilcoxon Rank Sum Test, all p < 0.001), check detailed statistics in AA19 of Table 3), indicating the spatial configuration of Cfix greatly reduced participants’ paying ordinary attention to each other.

Interviews

Post-task interviews with participants revealed more reflective insights into the spatial configurations. Around 41,000 words of transcription were transcribed and a thematic analysis of the transcription was undertaken. For more information about the thematic analysis, see (Braun & Clarke, 2006; Yin, 2017). The starting point of the thematic analysis was a reading through of the transcripts, then we did an inductive analysis of the data, collapsing relevant patterns into codes. Next, these codes were combined into overarching themes, which were then reviewed and adjusted until they fit codes well. As shown in Fig. 9, in total, 650 coded segments, 24 codes and 3 overarching themes emerged from the thematic analysis:

Figure 9 Ingredients of all the coded-segments of the interview, numbers of coded segments are shown along the bars.

.

Learning effects

Members of 18 groups mentioned the effect of the session sequence. Specifically, 40 coded segments contributed by 24 participants were related to learning effects. Participants reported the sequence is an “important factor” (Participant 15A, hereafter abbreviated to P15A). The first session was felt to be hard as they were “just being introduced to [the system and they were] still adjusting” to it (P5A), trying to “[figure] out how the system was working” (P16A). When they “were progressing into latter sessions, [they] felt easier to communicate and use gestures to manipulate the sound, being able to collaborate more, more used to the system” (P5B), these changes led to a higher level of satisfaction and more enjoyment in later conditions. It should also be noted that interestingly P11A and P11B reported the sequence effect adversely, they enjoyed the first session more because “the first one was an element of surprise, a total surprise” as that was “the first time they were using the system”. That feeling of freshness made that session more exploratory and more joyful to them. These learning effects might possibly affect the results of Post-Session Questionnaire and Comparison Questionnaire and thus should be well counter-balanced.

Reporting the spatial configurations

(1) Cpub - Simple but can be chaotic. Since there is no personal space, participants could, and had to hear all the interfaces all the time. In total, 16 coded-segments are about the disadvantage of this setting, some examples are: “a bit troubling” (P11B), “music [was] always very loud” (P9A), “it was global music, and there was someone annoying” (P2A), “[they were] not going to say anything” about disagreement because that could possibly make them appear to be “rude” (P2A). It was easier if there is something helpful “to perceive what [they were] doing, and not get confused with what [their collaborator] was doing” (P15B), it was too “chaotic” (P20A), “too confusing” (P22A) & P22B), “annoying” (P25B). They “cannot concentrate” (P25B), “everything [was] open and quite noisy” (P26B), they didn’t “have the tranquillity to operating [the] sounds...everything [came] mixed, which [was] difficult to manage” (P22A).

There were 25 coded segments from 14 participants reporting the positive side of the Cpub, some examples are: (i) pieces created in “personal space” might clash in a music way (P1A), “better to work when knowing how it sounds all together” (P17B), music pieces might match better; (ii) better for providing help to the other, according to P4A, they needed someone to lead them and thus the ability to hear all the work all the time was helpful; (iii) “space wise”, namely, no space limitation, compared with having to work closer to “hear the sound well” (P12A), Cpub does not have this constraint, they could choose to work “anywhere” (P24A); (iv) “easier” to understand the condition (P6B), fewer confusions when simply being able to hear all the things all the time (P13A); (v) “collaborative wise” (P13A), less separation, better collaboration compared conditions where “personal space” was provided (P3B, P18A and P18B).

(2) Cpub - Simple but can be chaotic. There were 34 coded segments contributed by 24 participants favouring condition Caug, higher than 12 segments contributed by 11 participants for Cpub, 10 segments contributed by eight participants for Cfix, and 19 segments contributed by 17 participants for Cmov (the sum of number of contributors here is greater than 52 as a few participants reported more than one favourite condition in the interview). The reason for the popularity can be concluded from the overwhelming 111 coded segments from 33 participants from 25 groups reporting the advantages of this condition, much higher than the number of segments reporting other conditions’ advantages. Caug’s advantages reported by participants can be grouped into four groups: (i) Higher team cohesion and less sense of separation. Participants reported that without the rigid personal space, they had to “work with the other person” (P6A). With no rigid personal space, Caug “forced [them] to collaborate more...because [they] had to stay very close” to compose music (P9B). (ii) An appropriate environment for creativity, more consistency and convenience. As described by participants, it was “a middle point between personal space and no personal space” (P6A), without even triggering something, “[they] could decide in a continuous way”, “whether [they] were able to listen to the other sound sources or not, [and] to what extent [they] wanted to isolate [themselves]” (P16A). Compared with having to hear all sounds in Cpub, this provided them with a “less stressing” (P4A) context, and they could selectively move away to avoid “getting interrupted with the other” (P5B) and overlapping music. Compared with Cfix and Cmov, being able to still “hear a bit of it in the background but not completely” (P20A) was reported good as this kept them “up to date” (P9A) and helped them to “tailor what [he/she] was making” (P22B) to match the co-created music and to make something new and see if it “fit with” (P20A) the old. Caug provided them with “a little bit of personal space” although not a quite “defined thing” (P6A), which provided the possibility “to work on something individually” and to “share work quite easily” (P20A). (iii) Easier to identify sounds. Participants reported it was easier to “locate the source of the sound” (P16A) and “perceive what (they were) doing” (P15B), these factors then helped them “understand instruments better” (P7B) and “not get confused” (P15B); (iv) More real. Quite interestingly, instead of Cpub, which simulates the acoustic attenuation in the real world, Caug was reported to be similar to the experience in the real world. Participants reported in Caug “if [they] want to hear something, [they] just come closer, like in the real world” (P11B & P11B), “feeling like the real-time experience (P26B)”.

It should also be noted that, along with these 111 coded segments reporting the advantages provided by Caug, there are 19 segments reporting its limitations. These limitations fall into three groups: (i) a preference “to hear all the instruments all the time” in Cpub (P26B), (ii) Caug might lead to “another type of compositions” and “influence the piece” (P16B), and (iii) not being able to hear all sounds led to a feeling of separation (P18A).

(3) Cfix and Cmov - resemblance and differences. Regardless of the mobility, the personal space provided by Cfix and Cmov share the same characteristics. Not surprisingly, the participants reported many common advantages and disadvantages shared by both conditions, including: The addition of rigid personal space was described as an “added advantage” (P7A), it made it “easier to perceive what [they were] doing and not get confused with what [their collaborator] was doing” (P15B), provided them with a chance to “isolate themselves to create their piece” (P22A), and to “think about something to add” (P9A), helped them to “develop their own ideas” (P8A). As a result, they used personal space “a lot [and] used [their] own creativity much more comparing with [other two sessions]” (P3A & P3B).

Common disadvantages reported include: The rigid form led to segmentation, and a feeling of being “forc[ed]” to work on something individually (P6A), making them “forget” the collaboration and collaborator (P8A & P12A), resulting in less collaboration, less “communication happening” (P7A), “lost the idea of the joint music piece” (P16A & P16B), and as a possible result, each other’s music pieces did not fit when brought up (P9A). P4A reported they were not familiar with music, and thus they “needed somebody to lead” them, so preferred to hear sounds all the time. Besides, P24B reported that the visual personal space made the stages look “messy”.

Differences between Cfix and Cmov—In total, 46 coded segments (from 26 participants) were reporting Cmov’s advantages and 26 segments (from 12 participants) reporting its disadvantages, compared with 22 segments (from 14 participants) and 56 segments (from 34 participants) for Cfix, indicating in general participants thought Cmov better than Cfix. Some example insights behind the preference are: Cmov functioned like a “mute button” (P4B), which could be used anywhere (P7B), enabling them to “move around”, work “closer...and see each other’s things” and thus led to “more collaboration” between them (P1B). Though Cfix had no advantages on these aspects, the location at the opposite corners provided a more “personal feeling” and a higher sense of belonging (P22A & P22B). Walking to the corner to access personal space was not a big issue for P7B & P7A as “the boundary is small”. Besides, the relatively far distance also helped to “prevent [them] from clashing” (P7A).

Discussion

The key question that this paper tries to address is how shared virtual environments should be designed to support creative collaboration. To help answer this question, a better understanding of the role of personal and public space is needed given that previous research has highlighted the necessity of providing personal space with fluid transition to public space during collaboration (Scott, Carpendale & Inkpen, 2004; Shen, Everitt & Ryall, 2003; Sugimoto, Hosoi & Hashizume, 2004) and also that people do form public and personal space in collaboration in VEs (Men & Bryan-Kinns, 2019). Next, based on the results, we will discuss (i) the necessity and (ii) the impacts of adding each type of personal space, then make comparisons between (iii) conditions providing personal space with and without mobility, and iv) conditions providing personal space with rigid and fluid boundary.

Necessity of adding personal space

When no personal space was available, Cpub was reported to provide the experience of the least difficulty for tracking the collaborator (CQ3 in Table 2), the strongest sense of collaborator (CQ4), best communication quality (CQ5), the least difficulty to cooperate (CQ9). So Cpub seems to be the simplest one among these four configurations for participants to learn and get used to. However, the issues of having no personal space are clear. Firstly, especially for the music making task in this study, according to the interview results, participants reported that the background could be messy to develop own ideas, their creativity requires a quieter and more controllable environment. Considering individual creativity forms an important part of the collaborative creativity, providing an appropriate environment is crucial. The personal space provided in Caug, Cfix, and Cmov functioned like a “less stressing” context, within which, participants could better “understand instruments” and not “get confused”. Secondly, participants need an opportunity to develop their own ideas. From the interview results, having personal space was reported to be “an added advantage”, it helped to promote their own creativity, which was then combined and contributed to the joint piece. This echoes the findings in Men & Bryan-Kinns, (2019), that providing personal spaces is helpful as it provides a chance to explore individual ideas freely, which then added an interesting dynamic to the collaborative work. Though some disadvantages of having personal space were also reported, e.g., less communication, higher isolation and being messy, most of these limitations were possibly the result of introducing rigid visible personal space, and Caug was founded to have addressed these limitations well (details will be discussed in later subsections). Next, we discuss the impacts of introducing each of these personal spaces individually by comparing their condition with Cpub.

Impacts of adding personal space

As mentioned above, the previous study (Men & Bryan-Kinns, 2019) found that the addition of personal space located at the opposite side of the public space led to a shrunken size of group territory, fewer group note edits, a larger size of personal territory, more personal note edits, a larger average distance between collaborators, and fewer times of paying attention to collaborators. We argued these negative impacts are mainly due to, in that study, the personal spaces being distributed on the opposite side of the group space resulted in a larger distance between participants. So we proposed personal space with different features (e.g., gradual boundary in Caug, mobility in Cmov) might reduce, or even minimise, these negative effects. Next, the impacts of introducing newly-featured personal spaces in Caug, Cfix, Cmov, and how these negative effects might be eliminated or minimised will be discussed.

Invisible auditory personal space in Caug

Caug is quite similar to Cpub in many ways, e.g., both do not have a visual boundary for personal spaces, no visual triggers for personal space, similar territorial patterns were formed in these two conditions (Fig. 8). Not surprisingly, no significant differences were found in most of the statistical measures listed in Tables 1 and 3. The only differences revealed in these two Tables are the significant differences found in AA16, AA17—significantly more occurrences and longer duration of close attention were paid to collaborator in Caug, and a marginal-significant difference in PSQ7—sense of collaborator’s activity is higher in Cpub than in Caug. From another perspective, fewer differences between Cpub and Caug indicate that the limitations of adding personal space identified in previous work (Men & Bryan-Kinns, 2019) have been successfully minimised. Specifically, the size of group territory and the number of group edits maintained similar numbers, the means of Caug are even greater, though not significantly (AA5 and AA7 in Table 3). Cpub and Caug saw a similar size of personal territory and personal edits, and similar average distance (respectively, AA6, AA8, AA10 in Table 3), and Caug even saw more close-attention paid to each other (AA16 and AA17 in Table 3). All these similarities indicate that by introducing a personal space with gradual and invisible boundary, these identified disadvantages of introducing personal space have been successfully eliminated. Reasons can be that Caug managed to provide a similar interaction experience to Cpub. In the previous study (Men & Bryan-Kinns, 2019), to access the personal spaces located at the opposite side of the public space, participants had to drift apart, which might have influenced the their spatial locations, changed the formation of group/personal territory they formed and the average distance between collaborators changed, and territoriality-based interaction (group/personal edits) changed. Here in Caug, by enabling participants to use personal space anywhere inside the stage with no specific triggers needed, Caug managed to provide a user experience as similar as possible to Cpub. The second reason is more related to the impacts on subjective experience, by making the personal space invisible and gradual, the isolation and difficulty of coordinating that introduced by the additional rigid personal space was minimised. For instance, in the interview, participants reported Caug provided a proper level of group work as a working context, making easier to create new that matches the old.

Movable personal space in Cmov

In Cmov, participants could pop up the personal space anywhere in the stage. In this way, personal spaces was provided with mobility. By doing so, several aforementioned negative effects found in Men & Bryan-Kinns (2019) were reduced. Specifically, these include the size of group territory, the average distance, times of paying attention to collaborator (respectively, AA5, AA10, AA16, and AA17 in Table 3). However, some significant differences remained, participants still had significantly fewer mutual note modifications, marginal-significantly fewer group edits and significantly more personal edits after personal space being introduced in Cfix and Cmov (respectively, AA4, AA7, and AA8 in Table 3). This can also be verified by the interview results. Compared with Caug, participants reported a higher sense of isolation in Cmov and Cfix, both providing rigid-form personal spaces. Namely, Cmov, by making the personal space available anywhere in the stage, managed to drag participants closer, saw a similar group territory, however, participants’ behaviour was still affected in many ways. Participants were still separated to some extent, which can be seen as a disadvantage of adding visible, solid personal space. In other words, Cmov did better than Cfix in minimising the negative impacts of adding personal space, but not as good as Caug.

A more rigid personal space in Cfix

Cfix provided a much more inflexible personal space, which influenced participants’ behaviour in many ways (see the significant differences between Cpub and Cfix in Tables 1 and 3). Not to mention participants’ polarised ratings on Cpub and Cfix in CQ3, CQ4, CQ5, CQ7 of Table 2: Significantly many participants reported the least difficulty of tracking collaborator’s activities (CQ3), the strongest sense of their collaborator’s presence (CQ4), the best communication quality (CQ5), the least difficulty of cooperating with collaborator (CQ7) happened in Cpub, whilst Cfix was thought conversely by significantly many participants. Their dislike of Cfix can also be seen in the interviews, in which the number of coded segments favouring Cfix and the number of segments reporting its advantages are the lowest, whist the number of segments disfavouring it and the number of segments reporting its disadvantages are the highest among the four conditions.

Providing personal space with fluid boundary

Measures in Table 3 show that Caug significantly differs from Cfix and Cmov in many ways. When both significant differences (p <0.05) and marginal-significant differences (p < 0.1) are considered, compared with Cfix and Cmov, Caug saw a smaller personal territory (AA6) and a bigger group territory (AA5), more mutual modifications (AA4), more group edits (AA7) and fewer personal edits (AA8), a larger distance between collaborators (AA10), more times of paying close attention (AA17) and a longer time of paying close attention (AA16). All these indicate that compared with the rigid personal space provided in Cfix and Cmov, the augmented acoustic attenuation in Caug enabled a closer collaboration, H3 is therefore supported. Caug’s advantages are shown in three ways, next, each will be specified.

Enough support for creativity with minimal impacts

PSQ2 (Table 1) questioned the support each condition gave to individual creativity. Although no significant differences were found, Caug has a higher mean rating, possibly indicating a higher level of support. It should be noted that all the questions in PSQ were phrased either positively (PSQ1, PSQ2, PSQ3, PSQ4, PSQ7) or neutrally (the rest), with no negative statements, which might have affected participants’ ratings positively. However, this imperfection has a limited influence on this study, because PSQ results are mostly used for comparisons between conditions, which are affected equally due to all of the conditions using the same phrasing. More insights regarding Caug’s helpfulness are revealed by the thematic analysis, according to which, Caug provided both “an appropriate background” with which participants felt “less stressed” and were able to “tailor” the individual composing to match the co-work, and a space personal enough to “work on something individually”. No major differences were found between Cpub and Caug, indicating Caug provides a very mild solution, with limited impacts on people’s collaborative behaviour being introduced, whilst still providing sufficient support for individual creativity during collaboration. Thus H2 is validated.

Closer collaboration and higher consistency

According to measures of attention (AA16, AA17, AA18, AA19 in Table 3), compared with other conditions, in Caug participants paid more close attention to their collaborator. Possible reasons for this can be found from the thematic analysis and other measures of Activity Assessments (Table 3). Compared with realistic acoustic attenuation in Cpub, Caug’s augmented acoustic attenuation setting forced or prompted people to work closer in order to hear each other’s work, as reported by some participants. Compared with adding personal space with visible rigid boundary, by enabling participants to “decide” whether to hear other’s work or not “in a continuous way”, an invisible gradual boundary in Caug led to less separation, and higher consistency between personal and public space, which matches the finding that people would like to be able to smoothly shift their artifacts from personal to public with intermediate shades in-between (Greenberg, Boyle & LaBerge, 1999). Compared with rigid personal spaces in Cfix and Cmov, Caug saw more mutual note modifications, more group note edits, and larger group territory, a closer average distance between collaborators (respectively, AA4, AA7, AA5, and AA10 in Table 3), all of these indicate that Caug saw a less separated collaboration than Cfix and Cmov. Compared with the three levels of privacy provided in UbiTable (Shen, Everitt & Ryall, 2003) and the binary levels of privacy provided in SharedNote (Greenberg, Boyle & LaBerge, 1999), the step-less sonic privacy provided by Caug in this study possibly managed to better echo the suggestion that a boundary between personal and public space should be provided with gradations in subtle and lightweight ways to enable a fluid shift (Greenberg, Boyle & LaBerge, 1999), H3 is therefore supported.

Popularity

The code “advantage of Caug” has 111 coded segments, which is far more than the segments other codes have. Thirty-five coded segments are “most favourite—Caug”, higher than all other three conditions. These indicate Caug is the most popular condition. This can also be partially verified by the preference measure. Specifically, Caug has the highest preference rating in PSQ3 (Table 1), and more participants chose Caug as the setting they most enjoyed in CQ1 (in Table 2). We believe the reasons behind this popularity are mainly due to its unique advantages, which as reported by participants, includes: (i) higher team cohesion and less sense of separation, (ii) an appropriate environment for creativity, (iii) easier to identify sounds and (iv) more real (though in fact, Cpub is more real from the perspective of simulation). These features of Caug made it provide better support for collaborative creativity and therefore led to its popularity.

Providing personal space without/with mobility

This subsection compares Cfix with Cmov. The clear, sole difference between these two conditions is the mobility of personal space. In Cfix, to access personal spaces at the corners, participants needed to physically walk away from the centre and head to the corner, which might be the reason that Cmov saw a closer average distance between collaborators than Cfix (AA10, Table 3). This greater distance in Cfix possibly resulted the significantly larger size of personal territories (AA6) and more personal edits (AA8) in Cfix. On the contrary, the closer distance in Cmov created more chances for participants to pay or draw attention between each other, as a result, significantly longer time was spent paying attention to collaborators (AA17, AA19 in Table 3). With a closer average distance and more attention paid to each other, participants reported they had a marginal-significantly better quality of communication in Cmov (PSQ6 of Table 1). On the other hand, with participants being far away from each other and less chances for contact in Cfix, significantly many participants reported that they had the worst communication quality in Cfix (CQ6, Table 2). Cmov was also rated by much fewer participants to be the least enjoyable condition than compared with Cfix (CQ1, Table 2). Besides, Cfix also led to a reduced sense of collaborator’s contribution (CQ9 in Table 3). As a possible result, Cmov saw a significantly more satisfying work output (PSQ5, Table 1).

The thematic analysis results also echo these findings. More coded segments are reporting Cmov’s advantages compared with those reporting Cfix’s, with more coded segments reporting Cfix’s disadvantages than those reporting Cmov’s. Also, more coded segments are favouring Cmov compared those favouring Cfix. Participants reported being able to use personal space anywhere in the stage with the personal space was good as it resulted in a closer distance, which led to more collaboration and made it possible to see each other’s work. To conclude, compared with Cfix, Cmov resulted in a better communication quality, produced better feeling of collaborator’s contributions, and was rated more enjoyable, thus it saw a closer collaboration and produced a more satisfying result, H1 is therefore supported.

Key Findings

In summary, the following are key findings from our results:

• Having personal space is suggested as it supports individual creativity, which is an important element of the collaborative creativity.

• Caug minimised the negative impacts introduced by adding personal space (previously identified by Men & Bryan-Kinns, 2019) better than Cfix and Cmov.

• Caug was found to have the most minimal impacts and even to influence the attention between collaborators positively. Both Cfix and Cmov produced a more alienated collaboration, indicators of which include significantly bigger personal territory and more personal edits, and significantly fewer mutual note modifications and fewer group edits, significantly lower sense of collaborator’s activity. Additionally, Cfix saw significantly more note edits, and significantly less ordinary attention paid between collaborators.

• Providing personal space with a fluid boundary is preferable, it provides enough support for individual creativity with the minimal cost, and can even lead to a closer collaboration (specifically, greater attention was paid between collaborators).

• Compared with stationary personal space, space with mobility led to better communication, produced a better feeling of collaborator’s contribution, had a higher rating in enjoyment, and produced a more satisfying output, and thus it supported collaboration better than stationary personal space.

Design implications

Based on the key points made above, we suggest five design implications for SVEs focusing on supporting creative collaboration (point 2 is for audio-related task only):

(1) SVEs supporting creative collaboration tasks should come with personal space, as it provides essential support for the development of individual creativity, which forms a key part of the collaborative creativity. This is especially essential when the output of the task is more disruptive (e.g., audio), co-workers need a space where they can think of and develop own mind and work.

(2) For audio-related tasks (e.g., collaborative music making), manipulating acoustic attenuation as personal space is an effective way to support both individual creativity and collaboration. It allows users to shift between personal and public working space continuously by adjusting their relative distance. It comes with light-weight form, functions as a personal space well, and can increase close attention paid between participants. Besides, based on our finding, it does not introduce significantly negative impacts whereas rigid personal space does.

(3) Beyond audio-related tasks, when providing personal space in SVEs, lightweight free-form personal space rather than personal space with rigid form should still be firstly considered, as it introduces fewer negative impacts on collaboration and enables a fluid shift, which matches the findings of Greenberg, Boyle & LaBerge (1999). The basis of providing light-weight privacy is not limited to audio, it can be provided by other modalitie(s) as well (e.g., visual). For example, in this study, augmented attenuation in sound has been verified to provide a useful personal space for CMM in SVEs. Similarly, a visual augmentation might be used for vision-related collaborative tasks (e.g., collaborative drawing) in SVEs. Multiple modalities can also be used simultaneously for tasks involving multiple modals, an example task can be making a short animation and creating an accompanying music track for it.

(4) Manipulating the level of augmentation (e.g., the augmented acoustic attenuation in this study) can change the level of feeling personal. In the Caug condition of this study, participants adjusted their distance between themselves and collaborators to obtain a different level of being personal (herein referred as “personalness”), e.g., total isolation can be achieved if both participants are working with a distance greater than 1.2 metres. We believe similarly, when personal spaces are provided with gradual and adjustable boundary, manipulating the parameter of the boundary (e.g., the degree of augmented attenuation) can impact the level of “personalness” and therefore adjust the impact of introducing personal space. For example, the augmented attenuation can be set to a very low level if an extremely minimal impact is being pursued. So adding a method enabling users to adjust the level can allow users to shift between having a “very personal” space with total isolation where they could not hear nor see each other’s work, and having no personal space when they have to work together. In this way, users can be enabled to manipulate the level between “personalness” and togetherness continuously, which is useful to allow users to develop own ideas and work together to tailor own work into the collaborative piece. Compared with adjusting “personalness” by distance in Caug, adjusting it by changing the parameter might also be useful as co-workers can then stay anywhere whilst still being able to adjust the “personalness” that the personal space provides.

(5) When it is hard or impossible to design a gradual, light-weight personal space that applicable to the task due to the type of the task, and a rigid-form personal has to be considered, it is better to provide rigid personal space with mobility, as the mobility feature gives users more freedom for accessing the personal spaces, and produces a better user experience with fewer negative impacts on the collaboration compared with rigid personal space without mobility. This implication also echoes the proposal raised in our previous work (Men & Bryan-Kinns, 2019).

Conclusions

In this article, we have briefed an experiment exploring how four different spatial configurations impact the collaboration differently. Both quantitative and qualitative data were demonstrated and analysed, comparisons between conditions were made, differences were found and five design implications were given. Specifically, augmented attenuation can support the individual creativity and the fluid shift between group activitiy and individual activity well during collaboration in SVE, with minimal negative impacts on collaboration introduced. We also found that a rigid personal space with mobility serves users’ needs better and is preferable over a non-mobile one.

In the future, we are keen to explore how to design and apply personal spaces with fluid boundaries in a wider range of creative scenarios in SVEs, e.g., for collaborative drawing in an SVE, personal space (visual privacy) might be provided by creating a foggy environment, the more far away from the drawing objects are, the more blurry the collaborators perceive them. We are also interested in how the boundary might be manipulated and whether the manipulation can result in different impacts on the collaborative behaviour.

Supplemental Information

Supplemental Information 1 A video about the system LeMo

Click here for additional data file.

Additional Information and Declarations

Competing Interests

Author Contributions

Ethics

Data Availability

1 Toybox Demo for Oculus Touch: https://www.youtube.com/watch?v=iFEMiyGMa58 (Accessed: 2019-09-03).

2 A PlayStation VR demo: https://www.theverge.com/2016/3/16/11246334/playstation-virtual-reality-social-vr-demo (Accessed: 2019-09-03).

3 Full source available at: https://sites.google.com/view/liangmen/projects/LeMo

4 The Queen Mary Research Ethics Committee granted ethical approval to carry out the study within its facilities (Ethical Application Ref: QMREC2005).

The authors declare there are no competing interests.

Liang Men conceived and designed the experiments, performed the experiments, analyzed the data, contributed reagents/materials/analysis tools, prepared figures and/or tables, performed the computation work, authored or reviewed drafts of the paper, approved the final draft of the manuscript submitted for review and publication.

Nick Bryan-Kinns conceived and designed the experiments, authored or reviewed drafts of the paper, approved the final draft of the manuscript submitted for review and publication.

Louise Bryce authored or reviewed drafts of the paper, analyzed the data, approved the final draft of the manuscript submitted for review and publication.

The following information was supplied relating to ethical approvals (i.e., approving body and any reference numbers):

The Queen Mary Research Ethics Committee granted ethical approval to carry out the study within its facilities (Ethical Application Ref: QMREC2005).

The following information was supplied regarding data availability:

The VR system of LeMo is available at: men, liang (2019): Unity Package for LeMo—A Multiplayer Music Making Prototype. figshare. Online resource. https://doi.org/10.6084/m9.figshare.9816395.v1.

Data is available at: Men, Liang (2019): Raw Data of Paper “Designing Virtual Spaces to Support Collaborative Creativity”. figshare. Dataset. https://doi.org/10.6084/m9.figshare.9820262.v1.

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
