# Peer review of "Designing spaces to support collaborative creativity in shared virtual environments"

_PeerJ Computer Science, doi:10.7717/peerj-cs.229_

## Round 0.1 · original submission · Minor Revisions

· Academic Editor

Minor Revisions

Both reviewers found your study to be important, valid and well presented. The reviewers made quite specific suggestions for "minor revisions" to the text and especially to figures. These revisions will make the paper clearer and more effective.

Please implement all of the suggested revisions or explain why you did not.

I look forward to seeing this paper published soon.

Reviewer 1 ·

Basic reporting

Although lengthy, the paper is well written with clear language and structure. It provides a thorough overview of the literature and introduces the reader into the subject matter of the experimental work. The paper introduces several hypotheses that are tested and the results are discussed and summarized at the end of the manuscript. The results are presented in a clear way for the most part. Some of the figure captions could be improved with additional context and explanations.

Experimental design

The work addresses important area of shared virtual experience in the context of sound which has been by and large neglected by the VR community.

The experimental design section needs to provide more information on the subjects. It is only mentioned in the abstract that 52 users were tested, while no specific information is given on the subjects' background, age, recruitment process.

Information on the experimental hardware setup should be provided: what VR headset was used, how was the tracking of the user position done, how were the hands tracked (input device).

The experiment includes four different conditions that the subjects experienced. Multitude of conditions poses threat to bias the subjects, however the authors address the learning effect by randomization and repeated use. The final analysis is performed on the last two sessions which seems adequate to address the issue of learning.

Validity of the findings

The results are discussed in details while the authors also provide summary of the results that and several key findings that could be used as guidance in designing collaborative spaces.

Additional comments

Captions for Figures should provide additional information on what results are presented.

Figure 4, it's not fully clear how this was presented tothe users. Could there be examples added for each what the users saw?

Figures 5 and 6 should include value of N (number of measurements). The chart labels for questions could perhaps also include the theme of the questions Q1-11, as in Table 2 (in parenthesis next to the question). Why are questions there labeled as CQ1-11 while labels in Figures are Q1-11?

Figure 7, caption should explain what the labels mean; location (shown as dots), direction (shown as arrows), etc. Why was this particular group selected?

Figure 8 needs more explanation. Is this top view? How was it measured?

Page 13, last paragraph.. "The redder/bluer..." should be rephrased to refer to the color intensity.

Reviewer 2 ·

Basic reporting

The authors present a shared virtual environment system called LeMo, which was developed to support collaborative music making between pairs of people. This study tested four boundary configurations of shared and personal space within the environment, with the goal of determining which configuration is preferred for creative collaboration by users. The authors build on their previous work in understanding personal space, or territory, within the virtual environment. In this paper, the territory is manipulated through an experimental design and tested with 26 pairs of participants.

Basic reporting was overall okay, but needs minor revisions. I would like the authors to address the following items:

Lines 102-104 need more clarification as to how the authors addressed the concern in their project, and how it differs (or not) from this study. Additional context would help here.

Line 106, define 'tabletop'.

Lines 171-172, address how the new system differs from the old one, and briefly why the changes were made. The authors also note that Figure 2 is from the other 2019 study, which is confusing to readers since it was stated that the new system is very different. Did the interface not change?

The results section could use clarifications. Please review and clarify the language in lines 350-414.

Figures 6 and 7 have corresponding colors to indicate conditions, but the distinction is not strong on black/white printouts. Please make it more clear without using color which condition is which.

Line 464: What is Cpi? I think this is a typo.

Experimental design

There were multiple areas where what appeared to be research questions were stated (lines 46-47 and 76-78). Please clarify which set of questions you sought to answer and ensure that they are the same question if repeated elsewhere in the paper.

The experimental design itself was fine with four independent variables. The authors accounted for ordering effects through counterbalancing. The questionnaire and interviews after seem standard.

Hypotheses were clear.

I do have a concern about the Post Session Questionnaire in Table 1. The questions were either phrased positively or neutrally, with no negative statements (e.g. The spatial configuration of this virtual world was difficult for me to understand). Please provide rationale for using positive or neutral phrases in the questionnaire.

The description of how the study was conducted was thorough. The statistics provided were sufficient to understand how each condition performed.

Interview transcription and analysis was clear. The theme on learning effects was especially useful seeing as this is a limitation of such work, so it was important to address this in the results.

Validity of the findings

Aside from the language clarifications requested above, this study was controlled and methods/procedures were implemented in a standard way.

The participant quotations brought the statistical findings to life and provided a deeper level of analysis to the themes that arose out of the questionnaire. Removal of the LeMo feedback (lines 618-621) kept the paper in focus.

There are minor grammatical errors in the Discussion section. I recommend a grammar check to rectify these errors.

The discussion related back to the results of this study and tied in with the results of the author's previous study of the same system. It would be helpful if the authors restated their research questions at the beginning of the discussion, and used them as references for the structure of the discussion.

Key findings are clearly stated and are based on the results of the study. Any speculation was presented using terms such as 'suggest' or 'seems'. Design implications are also based on the results and the discussion.

Overall the findings and design implications are interesting and of value for researchers working in the creativity field and/or the virtual environment space.

Additional comments

No additional comments.

---

## Round 0.2 · accepted · Accept

· Academic Editor

Accept

Thanks for your submission and for your work on the revision.

---

## Author Rebuttal · Round 0.2

Manuscript entitled "Designing Virtual Spaces to Support Collaborative Creativity."
The first revision

Response to Reviewers

12 August 2019

We really appreciate the reviews we received which will certainly help to improve our manuscript. We have addressed the points raised by the reviewers and would like to explain these changes below. The reviewers' comments are in bold text and our responses are in plain text.

**Response to Reviewer 1's Comments**

**The experimental design section … no specific information is given on the subjects' background, age, recruitment process.**
We have added additional information at the beginning of the experiment section. Specifically, the added information includes: the recruitment process (emailing group lists with the authors' school), participants' musical knowledge, computer skills, and the experience of using groupware and VR.

**Information on the experimental hardware setup should be provided: what VR headset was used, how was the tracking of the user position done, how were the hands tracked (input device).**
This information has been extended in the section "LeMo - An SVE for collaborative music making", see lines 184 - 187, and an in-text reference has been added when explaining the experimental procedure in line 324.

**Captions for Figures should provide additional information on what results are presented.**
Additional information has been added to the captions of these figures: Figure 4 has been remade, and in-picture captions have been added to explain the features of each condition; for Figure 5 & 6, the value of N (number of measurements) has been added; Figure 7's caption has been extended to better explain the visual elements of the figure.

**Figure 4, it's not fully clear how this was presented to the users. Could there be examples added for each what the users saw?**
Figure 4 has been remade, a first-person perspective showing all the elements in the virtual experimental scene and four third-person perspectives showing the four experimental conditions have been included. Third-person perspectives might make it easier to differentiate the difference between conditions.

**Figures 5 and 6 should include value of N (number of measurements). The chart labels for questions could perhaps also include the theme of the questions Q1-11, as in Table 2 (in parenthesis next to the question). Why are questions there labeled as CQ1-11 while labels in Figures are Q1-11?**
We have revised Figures 5 and 6 following this suggestion. Post-Session Questions (PSQ) in Table 1 are the questions shown in Figures 5 and 6, CQ (comparison questions) are another set of questions, asking participants to compare conditions at the very end of the experiment.

**Figure 7, caption should explain what the labels mean; location (shown as dots), direction (shown as arrows), etc. Why was this particular group selected?**
We have extended the caption to include the explanation. Group 3 was not chosen for any specific reason, the aim of Figure 7 is to show the visual trace in greater detail, so any group could have been

chosen. It also helps the reader make sense of Figure 8 (in which visual traces are much smaller). A brief explanation has been added in the text (in line 470-473), where Figure 7 is first mentioned.

**Figure 8 needs more explanation. Is this top view? How was it measured?**
We have extended the caption by explaining that these visual traces are top-view of visualised participants' locations, directions and musical note edits, and they are made based on system-logged data, including positions and rotations of participants' heads, and interactions with music interfaces, an in-text reference has also been added (471-473), referring data-log system to the sub-section "LeMo - An SVE for Collaborative Music Making", which explains how the data was gathered. The data-log system's explanation has been modified to be more detailed in line 207-211, e.g. the devices being used have been added.

**Page 13, last paragraph.. "The redder/bluer..." should be rephrased to refer to the color intensity.**
This sentence has been rephrased to "The more intensely blue or red the area is, the more presence the corresponding participant had shown in that location."

**Response to Reviewer 2's Comments**

**Lines 102-104 need more clarification as to how the authors addressed the concern in their project, and how it differs (or not) from this study. Additional context would help here.**
In lines 100-107, we have added more information about their projects and why their studies differ from ours: their study was performed based on 2D-screen media, which made their findings less informative for 3D media including VR.

**Line 106, define 'tabletop'.**
In lines 116-119, relevant references (such as Kruger et al., 2004, and Scott et al., 2004) have been added to define the term 'tabletop'.

**Lines 171-172, address how the new system differs from the old one, and briefly why the changes were made. The authors also note that Figure 2 is from the other 2019 study, which is confusing to readers since it was stated that the new system is very different. Did the interface not change?**
This study and the 2019 study are actually using the same, newer, LeMo system (2019), which was developed on the basis of the old LeMo system (2018). In lines 177-182, we have corrected the citation to make this point clearer. The old 2018 LeMo has a different interface, however, that study is not part of this paper except for introducing the system development history.

**The results section could use clarifications. Please review and clarify the language in lines 350-414.**
We have reviewed and clarified these paragraphs.

**Figures 6 and 7 have corresponding colors to indicate conditions, but the distinction is not strong on black/white printouts. Please make it more clear without using color which condition is which.**

We assume Reviewer 2 meant Figures 5 and 6 and so have strengthened the texture patterns in the bar charts (e.g. the dotted pattern has become thicker now, and grid pattern has been introduced).

We retained the colour hue because we believe this feature will ease the reading for people with normal vision without introducing any difficulties for people with visual impairments given the stronger patterns used.

**Line 464: What is Cpi? I think this is a typo.**
This is a typo, and it has been corrected.

**Experimental design**
**There were multiple areas where what appeared to be research questions were stated (lines 46-47 and 76-78). Please clarify which set of questions you sought to answer and ensure that they are the same question if repeated elsewhere in the paper.**
**The discussion related back to the results of this study and tied in with the results of the author's previous study of the same system. It would be helpful if the authors restated their research questions at the beginning of the discussion, and used them as references for the structure of the discussion.**
We have unified these questions into one overall question: How should Shared Virtual Environments be designed to support creative collaboration? This research question has now been restated at the start of the discussion, and followed by a brief structure of the discussion: ``...we will firstly discuss i) the necessity, ii) the impacts of adding each type of personal space, then compare iii) the differences of adding personal space with/without mobility, and iv) personal space with rigid/fluid boundary.''

**I do have a concern about the Post Session Questionnaire in Table 1. The questions were either phrased positively or neutrally, with no negative statements (e.g. The spatial configuration of this virtual world was difficult for me to understand). Please provide rationale for using positive or neutral phrases in the questionnaire.**
We agree that this is a limitation of the study, and we have stated this limitation and discussed the possible impacts of the results (e.g. participants might have given more positive ratings). However, this imperfection has a limited influence on this study, because PSQ results are mostly used for comparisons between conditions, which are affected equally by this imperfection due to all of the conditions using the same phrasing.

**The participant quotations brought the statistical findings to life and provided a deeper level of analysis to the themes that arose out of the questionnaire. Removal of the LeMo feedback (lines 618-621) kept the paper in focus.**
This section has been removed.

**There are minor grammatical errors in the Discussion section. I recommend a grammar check to rectify these errors.**
The discussion section has been proofread and grammatical errors have been corrected.

References:

Kruger, R., Carpendale, S., Scott, S.D. and Greenberg, S., 2004. Roles of orientation in tabletop collaboration: Comprehension, coordination and communication. *Computer Supported Cooperative Work (CSCW)*, *13*(5-6), pp.501-537.

Scott, S.D., Carpendale, M.S.T. and Inkpen, K.M., 2004, November. Territoriality in collaborative tabletop workspaces. In Proceedings of the 2004 ACM conference on Computer supported cooperative work (pp. 294-303). ACM.